# Therapeutic Senolysis of Axitinib-Induced Senescent Human Lung Cancer Cells

**DOI:** 10.3390/cancers16162782

**Published:** 2024-08-07

**Authors:** Hitoshi Kotani, Wei Han, Yuichi Iida, Ryosuke Tanino, Kazuaki Katakawa, Tamio Okimoto, Yukari Tsubata, Takeshi Isobe, Mamoru Harada

**Affiliations:** 1Department of Immunology, Faculty of Medicine, Shimane University, Izumo 693-8501, Shimane, Japan; kotani14@med.shimane-u.ac.jp (H.K.); hanbazi0515@yahoo.ac.jp (W.H.); yiida@med.shimane-u.ac.jp (Y.I.); 2Division of Medical Oncology & Respiratory Medicine, Department of Internal Medicine, Faculty of Medicine, Shimane University, Izumo 693-8501, Shimane, Japan; rtanino@med.shimane-u.ac.jp (R.T.); okimoto@med.shimane-u.ac.jp (T.O.); ytsubata@med.shimane-u.ac.jp (Y.T.); isobeti@med.shimane-u.ac.jp (T.I.); 3Department of Clinical Pharmacy, Faculty of Pharmaceutical Sciences, Shonan University of Medical Sciences, Yokohama 244-0806, Kanagawa, Japan; kazuaki.katakawa@sums.ac.jp

**Keywords:** axitinib, senescence, senolysis, lung cancer

## Abstract

**Simple Summary:**

Tyrosine kinase inhibitors (TKIs) inhibit receptor-mediated signals in cancer and vascular endothelial cells. Especially, axitinib inhibits signaling via vascular endothelial growth factor receptors (VEGFRs). In this study, we report an unforeknown effect of axitinib on human lung cancer cells. We show that axitinib increased the cell size and enhanced the expression of β-galactosidase in a panel of human cancer cell lines, irrespective of their expression of VEGFRs. Especially, axitinib-treated human lung adenocarcinoma A549 cells showed typical senescence and subsequent treatment with the senolytic drug ABT-263 induced drastic cell death (senolysis). Senolysis of senescent A549 cells by ABT-263 was attributed to caspase-dependent apoptosis and Bcl-xL inhibition. Reactive oxygen species were involved in axitinib-induced senescence, but not in senolysis, of A549 cells. In an A549-xenografted mouse model, combination therapy with axitinib and ABT-263 significantly suppressed tumor growth.

**Abstract:**

Background: Tyrosine kinase inhibitors (TKIs) inhibit receptor-mediated signals in cells. Axitinib is a TKI with high specificity for vascular endothelial growth factor receptors (VEGFRs). Aim: We determined whether axitinib could induce senescence in human cancer cells and be lysed by the senolytic drug ABT-263. Methods: Human lung and breast adenocarcinoma cell lines were used. These cells were cultured with axitinib or a multi-target TKI lenvatinib. The expression of β-galactosidase, VEGFRs, Ki-67, reactive oxygen species (ROS) of cancer cells, and their BrdU uptake were evaluated by flow cytometry. The mRNA expression of p21 and IL-8 was examined by quantitative PCR. The effects of TKIs on phosphorylation of Akt and Erk1/2, as downstream molecules of VEGFR signaling, were examined by immunoblot. The in vivo anti-cancer effect was examined using a xenograft mice model. Results: Axitinib, but not lenvatinib, induced cellular senescence (increased cell size and enhanced expression of β-galactosidase) in all adenocarcinoma cell lines. Axitinib-induced senescence was unrelated to the expression of VEGFRs on cancer cells. ROS were involved in axitinib-induced senescence. Axitinib-induced senescent lung adenocarcinoma A549 cells were drastically lysed by ABT-263. In A549-xenografted mice, combination therapy with axitinib and ABT-263 significantly suppressed tumor growth with the induction of apoptotic cancer cells.

## 1. Introduction

Tyrosine kinase inhibitors (TKIs) inhibit receptor-mediated signals in cells. TKIs inhibit cancer growth and tumor neovascularization, leading to tumor suppression [1]. Therefore, many TKIs have frequently been used in the treatment of various types of cancers [1]. In terms of the specificity of TKIs, some TKIs show broad specificity toward vascular endothelial growth factor receptors (VEGFRs), fibroblast growth factor receptor (FGFR), and platelet-derived growth factor receptor (PDGFR), called multi-targeted TKIs [2]. Such TKIs can exert effects on both cancer cells and tumor endothelial cells, and a representative is lenvatinib [2]. Alternatively, axitinib is an anti-angiogenic TKI that specifically targets VEGF-R1, VEGF-R2, and VEGF-R3 [3,4]. Therefore, axitinib has been used in the treatment of various types of cancers in order to target tumor vascular [1]. Clinically, the combination of immune checkpoint inhibitors with axitinib has been reported to improve the prognosis of patients with metastatic renal cell carcinoma compared with conventional therapy [5,6]. 

Recently, cellular senescence has received increasing research attention because various types of diseases have been associated with senescence [7]. Upon exposure to anti-cancer drugs, some cancer cells increased their cell size and survived by stopping or slowing their cell cycle in a DNA damage response involving increased expression of cell cycle inhibitors, including p16^Ink4a^ and p21^Cip1/Waf1^ [8,9,10]. Recent studies have suggested that cell cycle arrest in therapy-induced senescent (TIS) cancer cells is sometimes reversible, such that these cells could potentially re-enter the cell cycle and begin to grow again [11]. These TIS cancer cells are thought to be responsible for tumor progression and/or recurrence after anti-cancer therapy [12,13,14]. TIS cancer cells produce several inflammatory cytokines, such as IL-6 and IL-8, and tumor-promoting factors, called senescence-associated secretory phenotype (SASP) [15,16]. Thus, specifically targeting TIS cancer cells could lead to a novel and promising therapeutic strategy to prevent cancer recurrence [12,13]. Actually, a new approach to the selective removal of senescent cells, known as senolysis, has been proposed [17,18,19]. Senolytic drugs (senolytics) can preferentially induce cell death in senescent cancer cells [18,19]. Although ABT-263 (navitoclax) was developed as an inhibitor against Bcl-2, Bcl-xL, and Bcl-w to treat hematological malignancy [20], this reagent has been revealed to work as a senolytic drug [21]. Recently, we reported that ABT-263 can effectively lyse CDK4/6 inhibitor-induced senescent human breast cancer cells [22] and pemetrexed-induced senescent human lung cancer cells [23]. 

Although VEGFRs are mainly expressed on endothelial cells, some reports have shown that certain types of human cancer cells express VEGFRs [24,25]. This evidence suggests that axitinib could directly exert unknown effects on cancer cells. Therefore, in the present study, we investigated whether axitinib could induce senescence in four human lung cancer cell lines (three adenocarcinomas and a malignant pleural mesothelioma) and two human breast cancer cell lines. We demonstrate that axitinib, but not lenvatinib, can induce senescence in all human cancer cell lines tested independently of the expression of VEGFRs and intracellular VEGFR signaling. Specifically, using human adenocarcinoma A549 cells, we show that axitinib-induced senescence (increased cell size, enhanced expression of β-galactosidase, p21, and IL-8 mRNA) in A549 cells was reactive oxygen species (ROS)-dependent. In addition, axitinib-induced senescent A549 cells could be drastically lysed by the senolytic drug ABT-263 in vitro independently of ROS. Furthermore, the combination strategy of axitinib and ABT-263 was effective in an A549-xenografted nude mice model in vivo. Our findings reveal that axitinib could induce senescence in human lung adenocarcinoma cells and that this effect was unrelated to inhibition of VEGFR signaling of cancer cells, indicating an unforeknown effect of axitinib on cancer cells. The approach, consisting of axitinib-induced senescence followed by administration of a senolytic drug, could be a novel therapeutic strategy for preventing cancer recurrence.

## 2. Materials and Methods 

### 2.1. Cell Lines, Reagents, and Mouse Model

Three human lung adenocarcinoma cell lines (A549, PC9 and H1975) and a human malignant pleural mesothelioma cell line (H2452) were obtained from ATCC (Manassas, VA, USA). A549 has a wild-type p53 and a KRAS mutation, and PC9 has a p53 mutation (R248Q) and an epidermal growth factor receptor (EGFR) mutation. The cells were maintained in RPMI-1640 medium (Wako, Richmond, VA, USA) supplemented with 10% fetal bovine serum (Invitrogen, Waltham, MA, USA) and 20 µg/mL gentamicin (Sigma-Aldrich, St. Louis, MO, USA). They were grown at 37 °C in a humidified atmosphere with 5% CO_2_. Two human breast cancer cell lines (MDA-MB-231 and MCF-7), which were kindly provided by Dr. K. Takenaga, Faculty of Medicine, Shimane University, Japan, were maintained in DMEM (Sigma-Aldrich) supplemented with 10% fetal bovine serum and 20 µg/mL gentamicin. Axitinib and lenvatinib were purchased from ChemieTek (Indianapolis, IN, USA) and Selleck Chemicals (Houston, TX, USA), respectively. ABT-263 was purchased from Active Biochemicals (Hong Kong, China). ABT-199 and H1331852 were purchased from Selleck Chemicals and Cayman Chemical (Ann Arbor, MI, USA), respectively. The pan-caspase inhibitor zVAD-FMK was purchased from Enzo Life Sciences (Farmingdale, NY, USA). Necrostatin-1 and ferrostatin-1 were purchased from Santa Cruz Biotechnology (Santa Cruz, CA, USA). *N*-acetyl-l-cysteine (NAC) was purchased from Nacalai Tesque (Kyoto, Japan). Female BALB nude mice were purchased from CLEA (Shizuoka, Japan). The experimental protocol was approved by the Committee on the Ethics of Animal Experiments of the Shimane University Faculty of Medicine (IZ5-66, 74).

### 2.2. Cell Viability Assay

Cancer cells (2 × 10^4^ cells/well) were seeded in volumes of 100 μL into 96-well plates. After adding the indicated doses of reagents, cells were cultured for three days. Relative cell viability was evaluated using the Cell Counting Kit-8 (CCK-8; Nacalai Tesque, Kyoto, Japan). Then, 10 μL of WST-8 solution was added and the cells were incubated for an additional 3 h. The absorbances at 450 nm were measured using a microplate reader (Beckman Coulter). The relative cell viabilities (%) are shown.

### 2.3. Flow Cytometric Analysis

To examine the expression of VEGF-R1, R2, and R3 in cancer cells, cancer cells were stained with PE-conjugated anti-VEGF-R1, -R2, and -R3 antibodies. PE-conjugated anti-VEGF-R1 monoclonal antibodies were purchased from Miltenyi Biotec (Bergisch Gladbach, Germany) and PE-conjugated anti-VEGF-R2 and anti-VEGF-R3 monoclonal antibodies were purchased from BioLegend (San Diego, CA, USA). PE-conjugated mouse IgG was used as a control. Cell death was assessed using the annexin V-FITC Apoptosis Detection Kit (BioVision, San Francisco, CA, USA) and propidium iodide (PI). To examine β-galactosidase (β-gal) expression, cells were stained with SPiDER β-gal (Dojindo Molecular Technologies) for 30 min. The stained cells were analyzed using a CytoFLEX cytometer (Beckman Coulter, Brea, CA, USA). 

### 2.4. Measurement of Reactive Oxygen Species (ROS) 

Intracellular ROS were measured using carboxy-H_2_DCFDA (Molecular Probes, Eugene, OR, USA). Treated cells were cultured with carboxy-H_2_DCFDA (10 µM) for 30 min, and the collected cells were analyzed by flow cytometry.

### 2.5. Photography

Cancer cells were cultured in 6-well plates for 3 days with or without the indicated drugs. Photographs were taken using a Nikon ECLIPSE Ts2 (Nikon, Tokyo, Japan).

### 2.6. Quantitative Reverse-Transcription Polymerase Chain Reaction (qRT-PCR) 

The total RNAs of the cells were extracted using a PureLink RNA Mini kit (Thermo Fisher Scientific, Waltham, MA, USA). cDNAs were generated from the RNAs via reverse transcription using ReverTra Ace qPCR RT Master Mix with gDNA remover (Toyobo, Osaka, Japan). qPCR was performed using THUNDERBIRD SYBR qPCR Mix (Toyobo). The primers used for qPCR were as follows: 

p21 primers 5′-GGGACAGCAGAGGAAGAC-3′ and

5′-TGGAGTGGTAGAAATCTGTCA-3′; 

IL-8 primers 5′-GCCAACACAGAAATTATTGTAAAG-3′ and

5′-TTATGAATTCTCAGCCCTCTTC-3′;

β-actin primers 5′-CACCATTGGCAATGAGCGGTTC-3′ and

5′-AGGTCTTTGCGGATGTCCACGT-3′. 

The mRNA expression levels were subsequently normalized relative to β-actin mRNA levels and calculated according to the 2^−ΔΔCt^ method.

### 2.7. Cell Proliferation Assay 

Cell proliferation was evaluated by two different assays. Cells were cultured with axitinib or lenvatinib (2.5 µM). On day 3, the cells were harvested, incubated with BrdU (10 µM) for 3 h, fixed, and incubated with DNase I. Thereafter, these cells were stained APC-conjugated anti-BrdU antibody and analyzed by flow cytometry. On the other hand, cells were cultured similarly and fixed by cold 70% ethanol. The fixed cells were stained with APC-conjugated anti-Ki-67 antibody and analyzed by flow cytometry. 

### 2.8. Immunoblot Assay 

Immunoblot assay was performed, as previously reported [26]. The following antibodies were used as primary antibodies: anti-Erk1/2 (#4695; Cell Signaling Technology [CST], Danvers, MA, USA), anti-phospho-Erk1/2 (Thr^202^/Tyr^204^) (#4370; CST), anti-γH2AX (Ser^139^) (CR55T33; Invitrogen), anti-Akt (sc-8312; Santa cruz biotechnology), and anti-phospho-Akt (Ser^473^) (#4060; CST). Either goat anti-rabbit or horse anti-mouse horseradish peroxidase-conjugated antibody (#7074; CST, #7076; CST) was used as a second antibody. To detect β-actin, peroxidase-conjugated anti-β-actin antibody (#017-24573; FUJIFILM Wako Pure Chemical) was used. Protein bands were visualized using an Amersham™ ImageQuant™ 800 (Global Life Sciences Technologies Japan, Tokyo, Japan). The band intensities were scanned and quantified using the ImageJ software (http://rsb.info.nih.gov/ij/, accessed on 28 March 2024). 

### 2.9. In Vivo Mouse Model

To examine the antitumor effects of axitinib and/or ABT-263 on the in vivo growth of A549 cells, female BALB nude mice were injected subcutaneously with A549 (2.5 × 10^6^) cells and Matrigel (BD Biosciences, Franklin Lakes, NJ, USA) at a 1:1 volumetric ratio in a 100 μL solution. At 8 days after injection, the mice were divided into four groups. Axitinib (30 mg/kg) was administered orally on days 8, 9, 11, 12, 14, and 15. On days 10, 13, and 16, cancer-bearing mice were intraperitoneally injected with ABT-263 (25 mg/kg). Tumor volume (mm^3^) and body weight were measured twice weekly. The tumor volume was calculated as follows: tumor volume = (length × width^2^) ÷ 2. 

### 2.10. Apoptosis Detection in Tumor Tissue

The collected tissues were embedded in Tissue-Tek optimal cutting temperature compound (Sakura Finetek, Tokyo, Japan) and sliced into 10 μm sections using a Leica CM1520 cryostat (Leica Biosystems, Nussloch, Germany). A terminal deoxynucleotidyl transferase dUTP nick-end labeling (TUNEL) assay was performed using the Click-iT Plus TUNEL Assay Kit (Invitrogen, Waltham, MA, USA). Fluorescence images were acquired using a confocal laser scanning microscope FV3000 (Olympus, Tokyo, Japan).

### 2.11. Detection of Senescence-Associated β-Gal on Tumor Tissues

The embedded tissues were sliced into 12 μm sections. Fresh sections were washed twice with PBS and fixed for 15 min in 0.2% glutaraldehyde and 2% paraformaldehyde in PBS. The sections were washed twice with deionized water and incubated in 1 mg/mL X-gal (Promega, Madison, WI, USA), 5 mM potassium ferrocyanide, 5 mM potassium ferricyanide, 150 mM sodium chloride, and 2 mM magnesium chloride in citric acid/sodium phosphate buffer (pH 6.0) at 37 °C overnight. The sections were counterstained with a Nuclear Fast Red solution (ScyTek Laboratories, Logan, UT, USA).

### 2.12. Statistical Analyses

We used Student’s *t*-test for comparisons between two groups and analysis of variance (ANOVA) followed by the Tukey–Kramer test for comparisons among more than two groups for parametric analyses, with a significance threshold of *p* < 0.05.

## 3. Results 

### 3.1. Variable Sensitivity to TKIs among the Three Human Lung Cancer Cell Lines 

First, we examined the sensitivity of two human lung adenocarcinoma cell lines (A549 and PC-9) and a human malignant pleural mesothelioma cell line (H2452) to axitinib and lenavatinib. Axitinib is a TKI that specifically targets VEGFRs [3,4], whereas lenvatinib is a multi-target for TKIs [2]. The relative cell viability of A549 cells reached the bottom at a dose of 10 μM axitinib, whereas that of PC9 cells decreased dose-dependently until 40 μM axitinib. The IC50 of A549 was about 2.0 μM, but that of PC9 was about 5 μM. These results indicate that A549 and PC9 were highly and moderately sensitive to axitinib, respectively. Both cell lines were relatively resistant to lenvatinib. On the other hand, H2452 cells showed the similar sensitivity to both axitinib and lenvatinib (Figure 1A). The relative viability of the PC9 cells was increased when they were cultured with low doses (1.25–5 μM) of lenvatinib. A higher dose (40 μM) of lenvatinib decreased the viability of all cell lines. There was no definite difference in the effects of axitinib and lenvatinive lenvatinib on H2452 cells. Axitinib (2.5 μM) clearly increased the size of A549 cells without inducing cell death (Figure 1B). Thus, the antitumor effect of axitinib on these cells was cell arrest and not cell lysis. 

### 3.2. Axitinib-Induced Senescence in Human Cancer Cell Lines Independently of Their Expression of VEGFRs 

Microscopically, axitinib treatment increased cell sizes without any cell death. Given that cell enlargement is a feature of cellular senescence [7,8], we tested the possibility that axitinib could induce senescence in these cancer cells. When the three cell lines were treated with axitinib, forward scatter (FSC) levels increased, implying increased cell size (Figure 1C, upper). Interestingly, this cell enlargement was not observed in lenvatinib-treated cancer cells. Because β-gal expression is also a feature of cellular senescence [7,8], we investigated this possibility in TKI-treated cancer cells via SPiDER β-gal staining. Axitinib treatment led to increased β-gal expression in all cell lines, with the greatest increase observed in axitinib-treated A549 cells (Figure 1C, lower). Lenvatinib treatment increased β-gal expression in the PC9 and H2452 cells at low levels. 

Axitinib is a TKI with high specificity for VEGFRs [3]. Although VEGFRs are generally expressed on vascular endothelial cells, some cancer cells also express VEGFRs [24,25]. Therefore, we examined the surface expression of VEGFRs on the three cell lines. VEGF-R2 expression was mildly positive for all three cell lines, and A549 cells appeared to express a small amount of VEGF-R1 (Figure 1D). We also examined effects of axitinib and lenvatinib on phosphorylation of Akt and Erk1/2 as downstream molecules of VEGFR signaling [27,28]. In terms of A549 cells, both axitinib and lenvatinib inhibited phosphorylation of Akt, but promoted that of Erk1/2. Only axitinib inhibited phosphorylation of Akt of PC9 cells, and both axitinib and lenvatinib promoted phosphorylation of Erk1/2 of H2452 cells (Figure 1E and Appendix A). In addition, we determined whether TKIs could induce DNA damage response in cancer cells by examining the expression of γH2AX. Axitinib increased the expression of γH2AX in these cell lines at low levels. In contrast, such enhancement was not observed in lenvatinib-treated cancer cells. 

Additionally, we examined three additional human adenocarcinoma cell lines, including lung H1975, breast MDA-MB-231, and breast MCF-7. When the three cell lines were treated with axitinib, FSC levels and the expression of β-gal were increased (Figure 2A). These changes were not observed in lenvatinib-treated cancer cells. In addition, H1975 cells were lowly for VEGFRs, whereas MDA-MB-231 cells were negative for these receptors (Figure 2B). MCF-7 cells were highly positive for these receptors. Furthermore, axitinib decreased the expression of Akt only in MDA-MB-231, but promoted it in MCF-7. Axitinib promoted phosphorylation of Erk1/2 only in MCF-7 (Figure 2C). Axitinib marginally increased the expression of γH2AX only in MCF-7 cells. Totally, axitinib-induced senescence (increased cell size and increased expression of β-gal) in a panel of cancer cells lines was unrelated to the expression of VEGFRs and VEGFR signaling, indicating that axitinib induced senescence independently of its inhibitory effect on VEGFR signaling. 

### 3.3. Axitinib Induced SASP and Growth Arrest in A549 Cells 

Senescent cells acquire SASP features and show cycle arrest as a result of increased expression of the CDK inhibitors p16 and p21 [7,8]. Because the A549 cells in our previous study were negative for p16 and failed to produce IL-6 [26], we examined the effects of axitinib and lenvatinib on the mRNA expression of p21 and IL-8 in in vitro cultured A549 cells. We found that the axitinib treatment significantly increased the mRNA expression of both p21 and IL-8 (Figure 3A). We also validated effects of axitinib on cell proliferation by examining the expression of Ki-67 and uptake of BrdU. The results showed that axitinib, but not lenvatinib, apparently decreased the expression of Ki67 and the uptake of BrdU in A549 cells (Figure 3B). These results indicate that axitinib can induce typical senescence in A549 cells. 

### 3.4. Drastic Senolysis of Axitinib-Induced Senescent A549 Cells 

Recently, it was revealed that senescent cells could be effectively lysed by senolytic drugs [17,18] such as ABT-263 [20]. Therefore, we examined whether axitinib-induced senescent A549 cells could be effectively lysed by ABT-263. Untreated or axitinib-treated A549 cells were cultured with ABT-263 for 6, 12, or 24 h. As shown in Figure 4A, instances of drastic cell death were observed when axitinib-treated A549 cells were cultured with ABT-263 for 6 h, and the number of dead cells increased thereafter. In contrast, no dead cells were observed when untreated A549 cells were cultured with ABT-263 for 24 h. In the axitinib-treated A549 cells, the cell size increased and FSC levels shifted to the right. When flow cytometry was performed using whole cells and debris harvested from culture wells, the non-destroyed (ND) populations were drastically decreased in cells pre-treated with axitinib and subsequently cultured with ABT-263 for 6, 12, and 24 h (Figure 4B, upper). In terms of gating, annexin V^+^ or PI^+^ increased in ND cells that had been pre-treated with axitinib followed by subsequent ABT-263 for 24 h (Figure 4B, lower). These results are summarized in Figure 4C. Together, these data indicate that the estimation of ND populations by evaluating ABT-263-induced senolysis is more effective than examining the percentages of annexin V^+^ or PI^+^ cells after gating on ND cells. 

### 3.5. Senolysis of ABT-263 in Axitinib- or Lenvatinib-Treated Lung Cancer Cells

Next, we determined whether sequential treatment with axitinib or lenvatinib followed by ABT-263 could induce senolysis in three lung cancer cell lines. Axitinib treatment clearly increased the size of three cancer cells, whereas an effect of lenvatinib was not clear (Figure 1C), suggesting no ability of lenvatinib to induce senescence. Nevertheless, we investigated whether axitinib- or lenvatinib-treated three cancer cell lines could be lysed via ABT-263 treatment. In contrast to a drastic effect on the ND population of axitinb-treated A549 cells, ABT-263 showed no definite effect on the lenvatinib-treated A549 cells (Figure 5A). Compared to the experiments with A549 cells, the culture durations of PC9 and H2452 cells were prolonged because no definite cell death was observed after 12 h. When axitinib-pretreated PC9 cells were subsequently cultured with ABT-263 for 24 h, the ND population was significantly decreased by almost half, whereas the degree of senolysis was small in lenvatinib-treated PC9 cells. On the other hand, senolysis was not distinguishable in the axitinib-treated H2452 cells even when cultured for 72 h, and no senolysis was observed in lenvatinib-treated H2452 cells. These results are summarized in Figure 5B. These results indicate that no clear senolysis could be induced in lenvatinib-treated these cancer cell lines.

### 3.6. Senolysis of Axitinib-Induced Senescent A549 Cells Is Dependent on Apoptosis and Bcl-xL Inhibition 

Next, we investigated the mechanism underlying ABT-263-mediated senolysis in senescent A549 cells. First, we determined whether ROS were involved in the induction of senescence or excursion of senolysis. A549 cells exhibited increased ROS expression when cultured with axitinib, but not with lenvatinib (Figure 6A). A549 cells were treated with or without axitinib and NAC, a ROS scavenger, for 3 days, and ROS expression was examined. The addition of NAC hindered not only the ROS expression but also the enlargement of cell size and the β-gal expression partially (Figure 6B). Next, A549 cells were treated with axitinib for 3 days and, after harvesting, the cells were re-cultured with ABT-263 with or without NAC for 6 h. Then, cell death was examined by flow cytometry. As a result, ABT-263-mediated senolysis of senescent A549 cells was not inhibited by NAC (Figure 6C). These results indicate that ROS were involved in axitinib-induced senescence in A549 cells, but not in the execution of senolysis. 

In contrast, the addition of the pan-caspase inhibitor zVAD significantly inhibited the senolysis of axitinib-induced senescent A549 cells. This inhibitory effect was not induced by necrostatin-1 (a necroptosis inhibitor) or ferrostatin-1 (a ferroptosis inhibitor) (Figure 6D). Given that ABT-263 is an inhibitor specific to Bcl-2/-xL/-w [20], we examined the effects of either ABT-199 (a Bcl-2-specific inhibitor) [29] or A-1331852 (a Bcl-xL-specific inhibitor) on axitinib-treated senescent A549 cells. We found that although A-1331852 led to cell senolysis, as observed with the ABT-263 treatment, ABT-199 did not have this effect (Figure 6E). These results indicate that the senolysis of senescent A549 cells was dependent on caspase-dependent apoptosis and the ABT-263-mediated inhibition of Bcl-xL. 

### 3.7. Combined Effects of Axitinib and ABT-263 in A549-Xenografted Nude Mice 

Finally, we tested whether combined treatment with axitinib and ABT-263 could lead to antitumor effects in vivo using a A549-xenografted nude mouse model. Female BALB nude mice were injected subcutaneously with A549 cells and Matrigel. Axitinib monotherapy showed a (non-significant) tendency to suppress tumor growth, whereas ABT-263 monotherapy had no apparent effect. However, the combination of these agents significantly suppressed tumor growth in A549 cells on day 19 after tumor inoculation (Figure 7A,B). Images of tumor tissues from day 25 are shown in Figure 7C. On the other hand, although treatment with axitinib and/or ABT-263 significantly decreased the body weight of A549-bearing nude mice, this weight was recovered by 6 days after the last treatment (Figure 7D). TUNEL staining was performed using tumor tissues on day 17, the next day after the last treatment. In examining tumor tissues, we observed moderate and small amounts of TUNEL^+^ cells in the axitinib-treated group and the ABT-263-treated group, respectively. However, a substantial amount of TUNEL^+^ cells was observed in the combined treatment group (Figure 7E). We also examined the expression of β-gal^+^ cells in tumor tissues. Monotherapy with axitinib increased the number of β-gal^+^ cells, whereas the combined treatment of axitinib and ABT-263 decreased the β-gal^+^ cell population (Figure 7F). 

## 4. Discussion 

TKIs have been used for the treatment of various cancers [1]. In this study, we focused on a unique feature of axitinib, which is that it can induce senescence in human lung cancer cell lines. Axitinib has been reported to induce senescence in human renal carcinoma cells and glioma cells [30,31]. However, to our knowledge, no report has shown that axitinib can induce senescence in human lung adenocarcinoma and malignant pleural mesothelioma, as well as human breast adenocarcinoma. Furthermore, this is the first report of efficient senolysis induced by sequential treatment with axitinib followed by a senolytic agent. 

In this study, we showed that axitinib, but not lenvatinib, induced senescence in three lung adenocarcinomas and one malignant pleural mesothelioma. Because typical senescence was observed in A549 cells, we proceeded to analyze axitinib-induced senescent A549 cells. We found that ABT-263, which is a senolytic drug [21], effectively lysed axitinib-treated A549 cells. Although axitinib exhibits specificity to VEGFRs [3], and lenvatinib shows broader specificity toward VEGFRs, FGF, and PDGFR [2], we observed no senolysis in lenvatinib-treated A549 cells. We examined the expression of VEGFRs expression, phosphorylation of Akt and Erk1/2 as downstream molecules of VEGFR signaling [27,28], and the effects of TKIs on these molecules. The results were that the expression of VEGFRs and the effects of axitinib on phosphorylation of Akt and Erk1/2 were very varied among cancer cell lines. However, axitinib increased the cell sizes and the expression of SPiDER β-gal in all cancer cell lines tested (Figure 1C and Figure 2A). These results suggest that the inhibition of VEGFR signaling was not involved in axitinib-induced senescence. Although we have no clear answer why axitinib, but not lenvatinib, induced senescence, we have a hypothesis. It has been reported that ROS play important roles in axitinib-induced senescence [31,32] and axitinib induced higher levels of ROS than lenvatinib (Figure 6A). Interestingly, DNA-binding site-carrying anticancer drug bleomycin makes iron complex formation and ROS in the vicinity of DNA, resulting in strong DNA damage (cell death) [33,34]. On the other hand, axitinib makes two different iron complex formation and acts as a bidentate ligand to iron, whereas lenvatinib acts as a monodentate ligand to iron (Figure 8), explaining that axitinib can make higher levels of ROS than lenvatinib. However, in contrast to bleomycin, because axitinib has no DNA-binding site, axitinib cannot produce enough ROS in the vicinity of DNA, resulting in senescence as a result of insufficient induction of cell death. Although this scenario is a hypothesis, we would like to test this possibility in the next study. 

ABT-263 induced drastic cell death in axitinib-induced senescent A549 cells. Surprisingly, senolysis of senescent A549 cells was induced within 6 h after the ABT-263 treatment (Figure 4A,B). We performed flow cytometric analysis after annexin V/PI staining; however, the apoptosis assay was very difficult to interpret due to the extent of cell destruction (Figure 4C). Because enlarged senescent cells can be examined via FSC in flow cytometry, we used the percentage of ND cells to assess senolysis in A549 cells. We observed moderate ABT-263-mediated senolysis in PC9 cells and faint senolysis in H2452 cells (Figure 5A,B). What kind of roles does p53 play in the induction of axitinib-induced senescence and ABT-263-mediated senolysis? A549 and MCF-7 cell lines have wild-type p53, but PC9 and MDA-MB-231 cell lines carry mutant p53. Nevertheless, axitinib induced senescence in all cell lines. This means that p53 was unrelated to axitinib-induced senescence. Interestingly, it has been reported that SASP, a feature of senescence, is limited by wild-type p53 [35]. Alternatively, ABT-263-induced senolysis was higher in p53 wild-type A549 cells than in p53-mutated PC9 cells. At present, we suppose that p53 plays an important role in senolysis but not in the induction of senescence. On the other hand, in terms of the mechanisms of cell death, axitinib induced ROS in A549 cells (Figure 6A), and ROS was involved in the induction of senescence partially (Figure 6B). However, ROS was not involved in ABT-263-mediated senolysis of A549 cells. ROS are involved in aging-associated senescence [36], whereas these molecules contribute to various types of cell death [37]. We would like to elucidate roles of ROS in axitinib-treated cancer cells in more detail in the next study. 

Various types of cell death have been recently revealed. In addition to apoptosis and necrosis, necroptosis and ferroptosis have also been described [38,39,40]. Our experiments indicated that the drastic senolysis in the present study was due to caspase-dependent apoptosis. In addition, ROS was not involved in senolysis in A549 cells. The ABT-263-mediated senolysis was executed via Bcl-xL inhibition. Given that Bcl-xL plays roles in resisting cell death in various types of solid cancers [41], a sequential strategy consisting of axitinib followed by ABT-263 could be a novel and promising treatment for other types of solid cancers. 

In our xenograft mouse model, the combination of axitinib and ABT-263 significantly suppressed the in vivo growth of A549 cells (Figure 7A–C). In addition, combination therapy apparently increased the number of TUNEL^+^ cells in tumor tissues (Figure 7E). Furthermore, the ABT-263 treatment decreased β-gal^+^ cells in tumor tissues that increased via axitinib (Figure 7F). These results indicate that combination therapy with axitinib and subsequent ABT-263 holds promise. Although both axitinib and ABT-263 induced body weight loss when administered alone or in combination, this combined protocol would likely be tolerable because the body weight was regained within one week after the last treatment (Figure 7D). 

Senescent cells acquire SASP features and show cycle arrest as a result of increased expression of the CDK inhibitors p16 and p21 [7,8]. We measured the mRNA expression of p21 and IL-8 in in vitro cultured A549 cells that were treated with axitinib or lenvatinib (Figure 3A). Although we did not measure their mRNA expression in tumor tissues, we measured the expression of β-gal, as another feature of senescence, in A549 tissues after axitinib treatment in vivo, as shown in Figure 7E. However, this observation should be confirmed in human samples of axitinib-treated patients.

## 5. Conclusions

We demonstrated that axitinib can induce senescence in four human lung cancer cell lines and showed that senescent lung adenocarcinoma A549 cells could be drastically lysed by the senolytic drug ABT-263 in vitro. This combination strategy was also effective in an A549-xenografted mice model in vivo. There are several reports showing clinical application of axitinib for lung cancer patients in combination with chemotherapeutic drugs or immune checkpoint inhibitors [42,43,44,45]. However, to our knowledge, no report has shown axitinib-induced senescence and senolysis of human lung cancer cells. The approach, consisting of axitinib-induced senescence followed by administration of a senolytic drug, could be a novel therapeutic strategy for preventing cancer recurrence. 

## Figures and Tables

**Figure 1 cancers-16-02782-f001:**
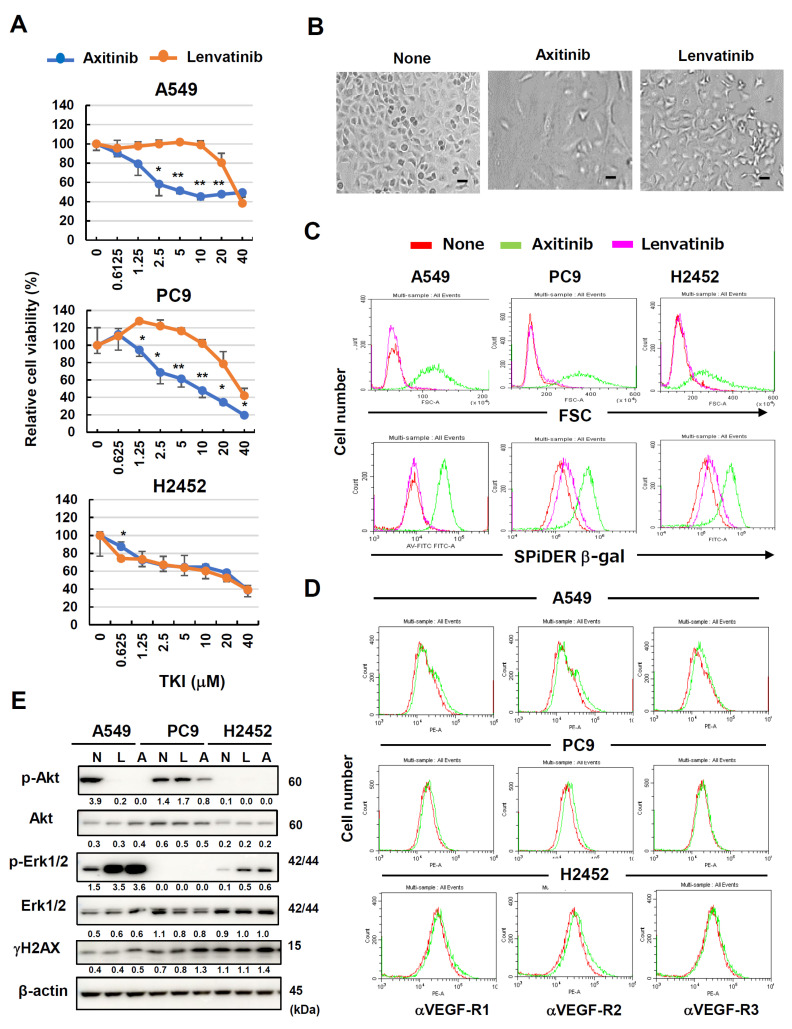
Effects of axitinib and lenvatinib on three human lung cancer cell lines. (**A**) Three lung cancer cell lines were cultured in the presence of indicated doses of two TKIs for 3 days. Relative cell viability was determined using a Cell Counting Kit-8 (CCK8) assay. Data are means ± SD of three replicates. * *p* < 0.05, ** *p* < 0.01. (**B**) Photographs of untreated A549 cells and those treated with axitinib or lenvatinib (2.5 μM) for 3 days. Scale bar, 50 μm. (**C**) Three lung cancer cell lines were cultured in the presence of two TKIs (2.5 μM) for 3 days. Thereafter, the cells were stained with SPiDER β-galactosidase (β-gal). Forward scatter (FSC) and SPiDER β-gal expression levels were examined. (**D**) Expression levels of VEGF-R1, -R2, and -R3 on three cancer cell lines were examined after staining with the indicated PE-conjugated antibodies (green line). Isotype-matched PE-conjugated antibodies were used as a control (red line). (**E**) Three lung cancer cell lines were cultured in the presence of two TKIs (2.5 μM). On day 3, the protein expression was examined by immunoblotting. β-Actin was used as a control. The numbers indicate the relative expression compared to β-actin. In the case with p-Akt and p-Erk1/2, the relative expression compared to relevant total protein is shown. The original Western blot figures is shown in Appendix A. N, none; L, lenvatinib; A, axitinib.

**Figure 2 cancers-16-02782-f002:**
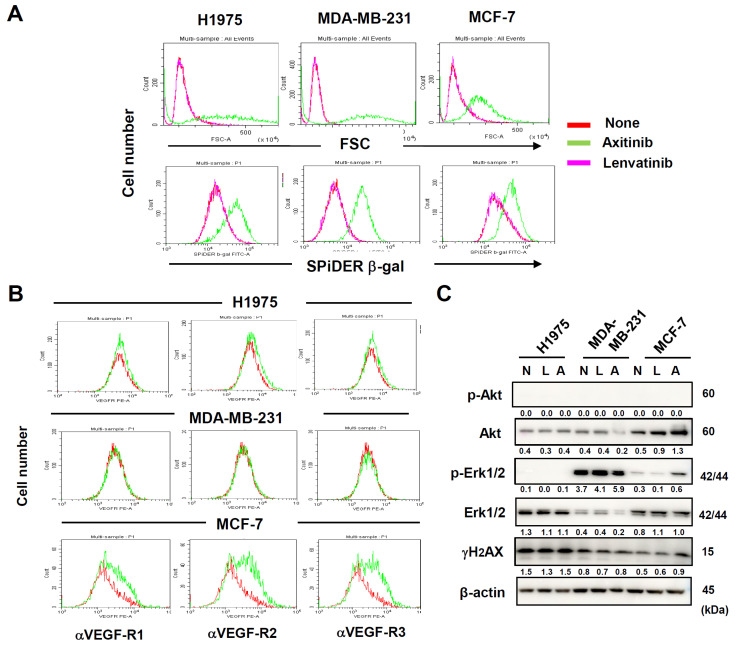
Axitinib-induced senescence in three other human adenocarcinoma cell lines. (**A**) Three cancer cell lines were cultured in the presence of two TKIs (2.5 μM) for 3 days and were stained with SPiDER β-galactosidase (β-gal). Forward scatter (FSC) and SPiDER β-gal expression levels are shown. (**B**) Expression levels of VEGFRs on three cancer cell lines were examined after staining with the indicated PE-conjugated antibodies (green line). Isotype-matched PE-conjugated antibody was used as a control (red line). (**C**) Three cancer cell lines were cultured in the presence of two TKIs (2.5 μM). On day 3, the protein expression was examined by immunoblotting. β-Actin was used as a control. The numbers indicate the relative expression compared to β-actin. In the case with p-Akt and p-Erk1/2, the relative expression compared to relevant total protein is shown. The original Western blot figures is shown in Appendix A. N, none; L, lenvatinib; A, axitinib.

**Figure 3 cancers-16-02782-f003:**
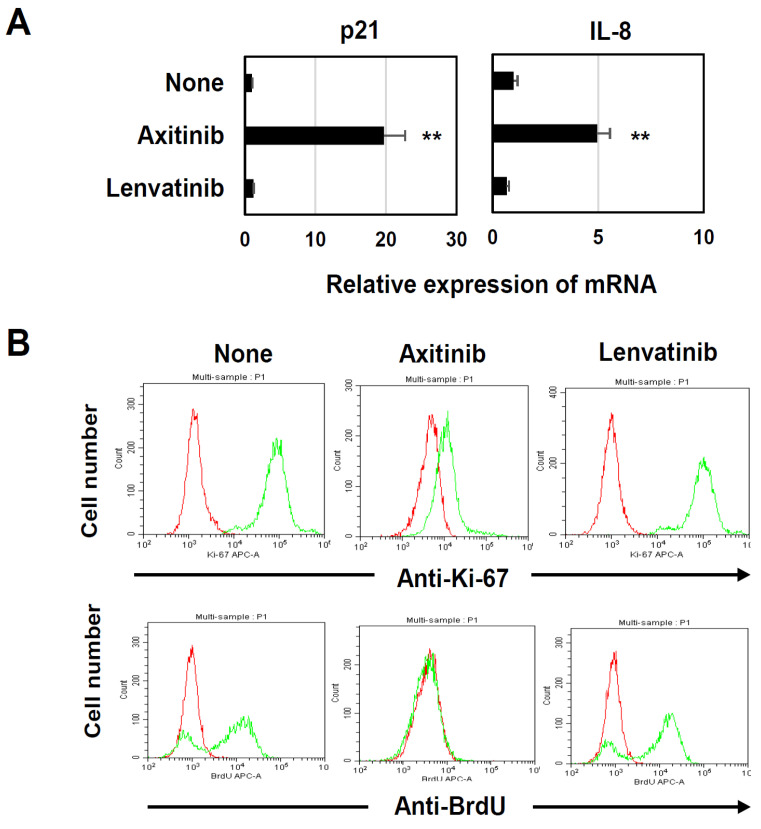
Induction of SASP and growth arrest in axitibin-treated A549 cells. (**A**) Three lung cancer cell lines were cultured with either axitinib or lenvatinib (2.5 μM) for 3 days. Expression levels of p21 and IL-8 mRNAs were determined in the harvested cells. Data are means ± SD of three samples. Means were compared using Student’s *t*-test (** *p* < 0.01 compared to the untreated control). (**B**) A549 cells were cultured with axitinib or lenvatinib (2.5 μM) for 3 days. Then, these cells were stained with APC-conjugated anti-Ki-67 antibody or APC-conjugated anti-BrdU antibody (green line) and analyzed by flow cytometry. The red line is an isotype-matched control antibody.

**Figure 4 cancers-16-02782-f004:**
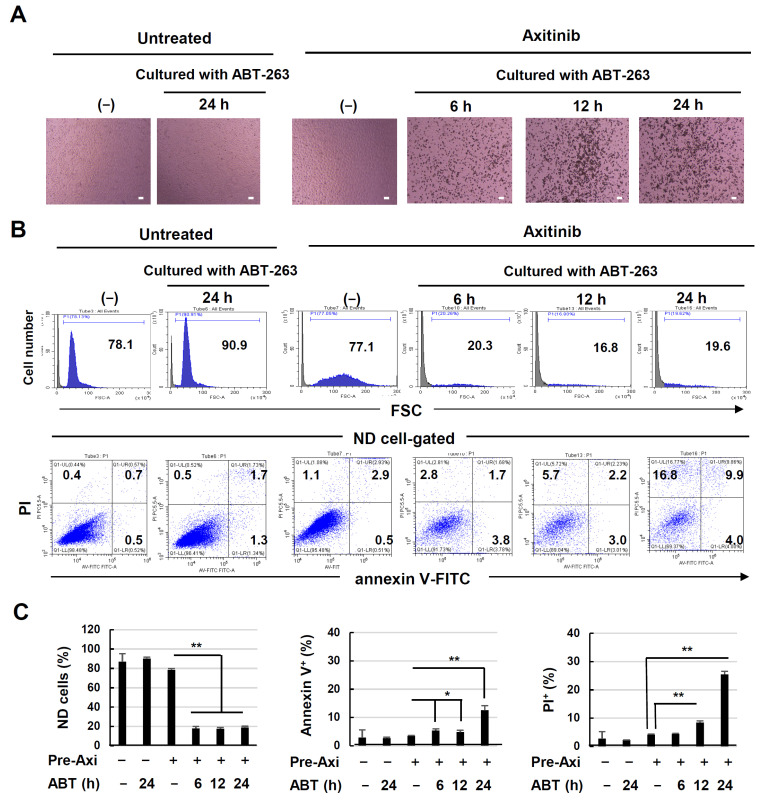
Senolysis of axitinib-induced senescent A549 cells by ABT-263. (**A**) A549 cells were treated with or without axitinib (2.5 μM) for 3 days. After harvesting, the cells were cultured with ABT-263 (1 μM) for the indicated times, and images were captured. Scale bar, 100 μm. (**B**) Flow cytometry was performed after staining with annexin V-FITC and propidium iodide (PI). Forward scatter (FSC) data are shown in the upper panel. The data gated on non-destroyed (ND) cells are shown in the lower panel. Numbers are percentages for each subset. (**C**) Percentages of ND cells, annexin V^+^ cells, and PI^+^ cells. Data are means ± SD of three replicates. Means were compared using Student’s *t*-test (* *p* < 0.05, ** *p* < 0.01).

**Figure 5 cancers-16-02782-f005:**
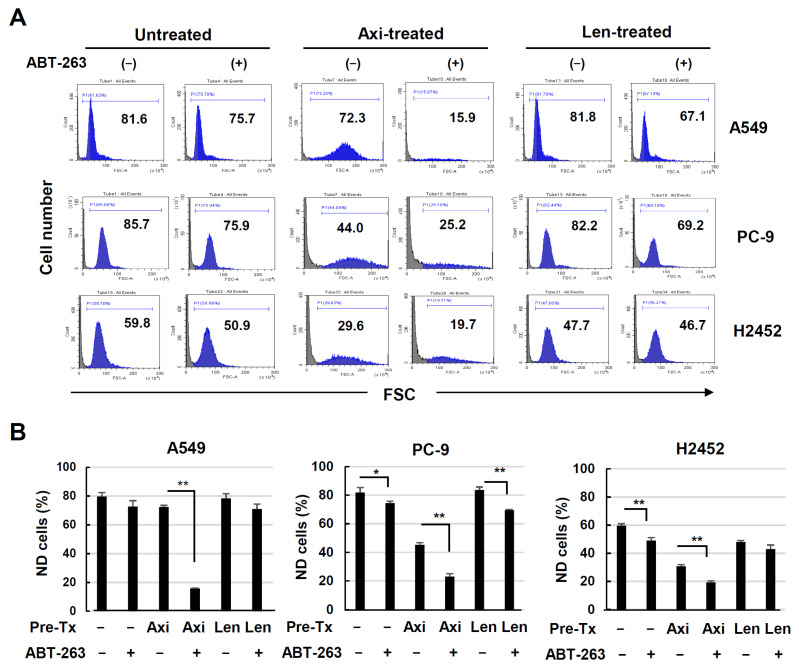
Senolysis of axitinib- or lenvatinib-treated three lung cancer cell lines. (**A**) Three lung cancer cell lines were treated with or without axitinib or lenvatinib (2.5 μM) for 3 days. After harvesting, A549, PC9, and H2452 cells were cultured with ABT-263 (1 μM) for 12, 24, and 72 h, respectively. Then, flow cytometry was performed. Forward scatter (FSC) data are shown. Numbers are percentages for each subset. (**B**) Percentages of non-destroyed (ND) cells. Data are means ± SD of three replicates. Means were compared using Student’s *t*-test (* *p* < 0.05, ** *p* < 0.01). Ax, axitinib; Len, lenvatinib.

**Figure 6 cancers-16-02782-f006:**
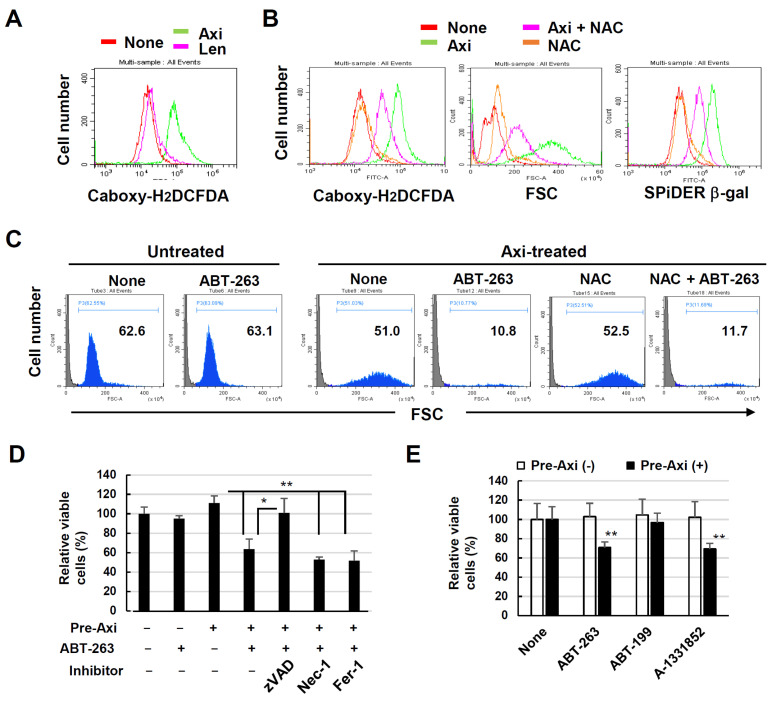
Senolysis of axitinib-induced senescent A549 cells by ABT-263 is dependent on caspase activation and Bcl-xL inhibition. (**A**) A549 cells were treated with or without axitinib (2.5 μM) for 3 days, at which point reactive oxygen species (ROS) expression was examined via flow cytometry after staining with carboxy-H_2_DCFDA. (**B**) A549 cells were treated with axitinib (2.5 μM) with or without NAC (2.5 mM) for 3 days. Then, the cells were stained with SPiDER β-gal. Forward scatter (FSC) and SPiDER β-gal expression data are shown. (**C**) A549 cells were treated with axitinib (2.5 μM) for 3 days. After harvesting, the cells were cultured with ABT-263 (1 μM) with or without NAC (2.5 mM) for 6 h. Then, flow cytometry was performed. FSC data are shown. (**D**) A549 cells were treated with or without axitinib (2.5 μM) for 3 days. After harvesting, the cells were cultured with ABT-263 (1 μM) for 12 h. Relative viability was determined via a CCK-8 assay. At 1 h before culture with ABT-263, either zVAD, Nec-1, or Fer-1 was added to the culture. (**E**) A549 cells were treated with or without axitinib (2.5 μM) for 3 days. After harvesting, the cells were cultured with ABT-263, ABT-199, or A-1331851 (1 μM) for 12 h. Relative viability of the cells was determined using a CCK-8 assay. Data are means ± SD of three replicates. Means were compared using Student’s *t*-test (* *p* < 0.05, ** *p* < 0.01). Nec-1, necrostatin-1; Fer-1, ferrostatin-1.

**Figure 7 cancers-16-02782-f007:**
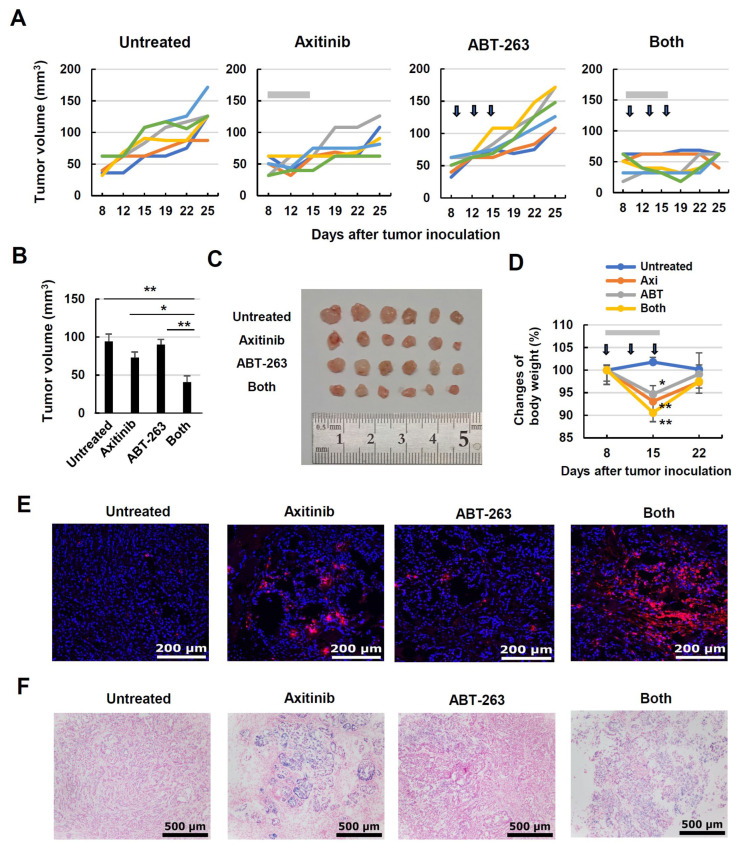
In vivo antitumor effect of axitinib and ABT-263 on the growth of A549 cells. (**A**) Female BALB nude mice were injected with A549 cells (2 × 10^6^) and Matrigel at a 1:1 volumetric ratio in 100 μL in the right flank. On day 8, the mice were divided into four groups. Axitinib (30 mg/kg; gray bars) was administered orally on days 8, 9, 11, 12, 14, and 15. On days 10, 13, and 16, cancer-bearing mice were intraperitoneally injected with ABT-263 (25 mg/kg; arrows). Tumor volume (mm^3^) and body weight were measured twice weekly. Lines represent the growth of individual mice (n = 6). (**B**) Tumor volumes on day 19. Data are means ± standard error on the mean (SEM) for six mice. Means were compared using analysis of variance (ANOVA) followed by the Tukey–Kramer test (* *p* < 0.05, ** *p* < 0.01). (**C**) Photographs (on day 25) of tumor tissues. (**D**) Body weight was measured twice weekly. Means were compared using ANOVA followed by the Tukey–Kramer test (* *p* < 0.05, ** *p* < 0.01). (**E**) Terminal deoxynucleotidyl transferase dUTP nick-end labeling (TUNEL) staining was performed using tumor tissues on day 17. Red dots indicate TUNEL^+^ cells. Scale bar, 200 μm. (**F**) Similarly, β-galactosidase staining was performed. Blue dots indicate β-galactosidase^+^ cells. Scale bar, 500 μm.

**Figure 8 cancers-16-02782-f008:**
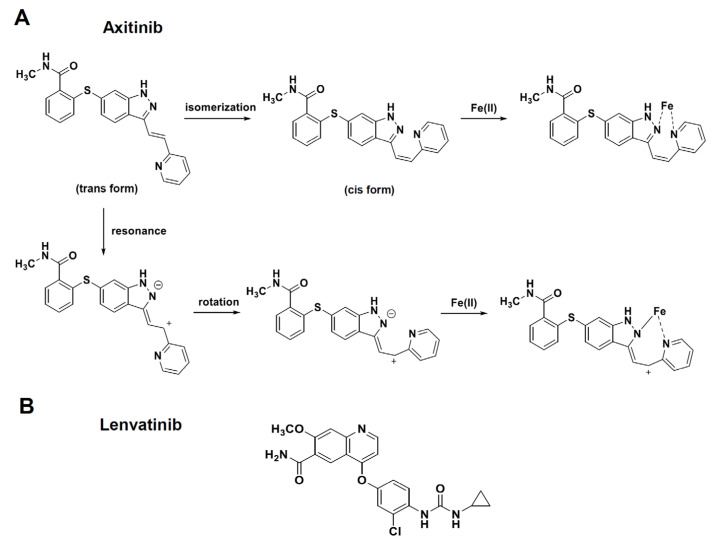
Speculated iron complex formations of axitinib and lenvatinib. (**A**) Axitinib makes two different iron complex formations via either isomerization or resonance and rotation. As a result, axitinib can act as a bidentate ligand to iron. (**B**) Lenvatinib acts as a monodentate ligand to iron.

## Data Availability

The raw data supporting the conclusions of this article will be made available by the authors on request.

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
