# Peer review of "Therapeutic Senolysis of Axitinib-Induced Senescent Human Lung Cancer Cells"

_cancers, 2024, doi:10.3390/cancers16162782_

Round 1
Reviewer 1 Report
Comments and Suggestions for Authors
This is a nice and comprehensive work about TKIs effects on cancer models, but the authors should perform some modifications before the decision.
1) The authors should revise the abstract by adding some obtained results. We prefer numerical results. The current abstract is only general facts about axitinib, and senolysis of A549 cells. Some parts of the text are confusing: E.g. What does it mean in lines 28-30? Abstract also has to be reformed logically: 2-3 lines background, 2-3 lines aim of the work, 2-4 lines method of the used and 6-10 lines obtained results from the work.
2) I strongly recommend the authors provide a graphical abstract at the end of the introduction.
3) The novelty and application of this work should also be mentioned in the introduction.
4) How did the authors mention the sensitivity of A549 and PC9? They claimed these cell lines were highly and moderately sensitive to axitinib (lines 182-183) since the three cell lines had similar behavior at 40 uM and the sensitivity/resistance of the cell line to axitinib should be explained in section 3.1.
5) Did the authors use any cytotoxicity or cell viability test, such as MTT or Alumar blue?
6) The property of axitinib should be described in the introduction. The relationship between TKI axitinib and lenvatinib must also be explained in the results.
7) The authors wrote in lines 258-259 “We found that the axitinib treatment significantly increased the mRNA expression of p21 and IL-8”. Did they measure the mRNA expression in cell culture or in vivo models? (This should be explained in the manuscript). If it is measured from cell culture models, how do they extend the results to animal or human bodies?
8) How did the authors make A549-xenografted nude mice? (should be explained in the manuscript). The results may be different from naturally cancerous mice. (cancer changes genetic and epigenetic regulation pathways)
9) I recommend Figure 8 should be moved to the introduction and explain about chemical structure and behavior of these compounds there.
10) Did the authors study or collect the data about pharmacokinetics of axitinib since they carried out of in-vivo experiment?
Author Response
Reviewer 1
Thank you very much for taking the time to review our manuscript. Please find the detailed responses below and the corresponding revisions/corrections highlighted (in red) in the re-submitted files.
1) The authors should revise the abstract by adding some obtained results. We prefer numerical results. The current abstract is only general facts about axitinib, and senolysis of A549 cells. Some parts of the text are confusing: E.g. What does it mean in lines 28-30? Abstract also has to be reformed logically: 2-3 lines background, 2-3 lines aim of the work, 2-4 lines method of the used and 6-10 lines obtained results from the work.
Response 1: Thank you for important comments. We are sorry that the Abstract was insufficient. According to your suggestion, we revised the abstract completely.
2) I strongly recommend the authors provide a graphical abstract at the end of the introduction.
Response 2: According to the suggestion, we prepared a graphical abstract and provide it at the end of the Introduction on page 4.
3) The novelty and application of this work should also be mentioned in the introduction.
Response 3: According to the suggestion, we mention the novelty and application of this work in the end of the introduction, page 2, line 87-91.
4) How did the authors mention the sensitivity of A549 and PC9? They claimed these cell lines were highly and moderately sensitive to axitinib (lines 182-183) since the three cell lines had similar behavior at 40 uM and the sensitivity/resistance of the cell line to axitinib should be explained in section 3.1.
Response 4: Thank you for important comments. We are sorry for unclear description. Both cell lines were decreased their cell viability at a dose of 40 uM axitinib. However, the relative cell viability of A549 cells reached the bottom at a dose of 10 uM axitinib, whereas that of PC9 cells decreased until a dose of 40 uM. The IC50 of A549 was about 2.0 uM but that of PC9 was about 5 uM. These results indicate that A549 and PC9 were highly and moderately sensitive to axitinib, respectively. We added these comments in section 3.1., on page 5, line 219 –224.
5) Did the authors use any cytotoxicity or cell viability test, such as MTT or Alumar blue?
Response 5: We apologize not for describing on the cell viability assay. We used the Cell Counting Kit-8 (CCK-8) assay. We describe this in section 2.2., on page 3, line 129-135.
6) The property of axitinib should be described in the introduction. The relationship between TKI axitinib and lenvatinib must also be explained in the results.
Response 6: According to the suggestion, we describe the property of axitinib in the introduction, on page 2, line 49-54. We also describe and the relationship between axitinib and lenvatinib in the results on page 5, line 219-220.
7) The authors wrote in lines 258-259 “We found that the axitinib treatment significantly increased the mRNA expression of p21 and IL-8”. Did they measure the mRNA expression in cell culture or in vivo models? (This should be explained in the manuscript). If it is measured from cell culture models, how do they extend the results to animal or human bodies?
Response 7: Thank you for insightful comments. We measured the mRNA expression in in vitro cultured cells. This is described on page 11, line 384. We examined the mRNA expression of p21 and IL-8 as a marker of SASP that is a feature of cellular senescence. On the other hand, as shown in Figure 7E, we measured the expression of beta-galactosidase, as another feature of senescence, A549 tissues after axitinib treatment in vivo. However, this observation should be confirmed in human samples of axitinib-treated patients. In the future, we would like to perform experiment using clinical samples. These are described in the Discussion on page 17, line 523 – 529.
8) How did the authors make A549-xenografted nude mice? (should be explained in the manuscript). The results may be different from naturally cancerous mice. (cancer changes genetic and epigenetic regulation pathways)
Response 8: The reviewer pointed out how A549-xenografted nude mice were prepared. We described it in the Material and the figure legend in the previous manuscript. According to the reviewer suggestion, we describe it in the manuscript on page 13, line 415-416.
9) I recommend Figure 8 should be moved to the introduction and explain about chemical structure and behavior of these compounds there.
Response 9: Thank you for insightful comments. However, we would like to show Figure 8 and discuss on it in the Discussion.
10) Did the authors study or collect the data about pharmacokinetics of axitinib since they carried out of in-vivo experiments?
Response 10: Thank you for important comments. To our regret, we did not study or collect the data of pharmacokinetics of axitinib.
Reviewer 2 Report
Comments and Suggestions for Authors
This manuscript characterized a tyrosine kinase inhibitor axitinib induced senescent cancer cell line model, and explored the potential synergetic anti-tumor effect of axitinib in combination with an anti-apoptotic BCL2 family inhibitor ABT-263. This is of some interest since treatment-induced senescence has implication to cancer recurrence. Some weaknesses include only one cell line (A549) for efficacy study, several mechanism gaps, no long-term xenograft treament study to assess recurrence free survival etc.
Besides, some specific points need to be clarified:
(A) Figure 6 seems be partly cropped and not presented completely. In Figure 6B, Axi+NAC decreased FSC compared to Axi, while in figure 6C, Axi+NAC didn't decrease FSC compared to Axi only?
(B) In Figure 7, it looks the mice were treated for 9 days, and sacrificed after another 9 days, what's the rationale for this design?
(C) Tissues shown in Figure 7C and Figure 7E were described as day 25 and day 17, are these different batches of mice? which batch generated the tumor volume curves?
Author Response
Reviewer 2
Thank you very much for taking the time to review our manuscript. Please find the detailed responses below and the corresponding revisions highlighted (in red) in the re-submitted files.
1) Figure 6 seems be partly cropped and not presented completely. In Figure 6B, Axi+NAC decreased FSC compared to Axi, while in figure 6C, Axi+NAC didn't decrease FSC compared to Axi only?
Response 1: Thank you for comments. The reviewer pointed out different in the expression levels of FSC of the Axi+NAC groups between Figure 6B and 6C. However, their experimental protocols were quite different. In Figure 6B, A549 cells were cultured with both Axi and NAC to examine an effect of NAC on Axi-induced senescence of A549 cells. On the other hand, in the Axi+NAC group of Figure 6C, A549 cells were cultured with Axi for 3 days. After harvesting, the cells were cultured with NAC with or without ABT-263 for 6 h to examine an effect of NAC on ABT-263-mediated senolysis of Axi-induced senescent A549 cells. Although these culture protocol had described in the legend of Figure 6, to avoid misunderstanding, we described these in the manuscript, on page 11, line 376 – page 12, line 381.
2) In Figure 7, it looks the mice were treated for 9 days, and sacrificed after another 9 days, what's the rationale for this design?
Response 2: Thank you for an important comment. Actually, there is no rationale for this protocol design. However, the therapeutic protocols are based on our previous experiments using A549-xenografted nude mice (Ref. 23). Because immunodeficient nude mice lack T cell acquired immunity, we judged that long-term observation of cancer-bearing nude mice was not necessary.
3) Tissues shown in Figure 7C and Figure 7E were described as day 25 and day 17, are these different batches of mice? which batch generated the tumor volume curves?
Response 3: Thank you for important comments. Experiments for Figure7C and Figure 7E were performed at different times. The experiment of Figure 7E was additionally performed after the experiment of Figure 7C. We supposed that the day 25, that is, 7 days after the last treatment, was too late to examine the expression of beta-galactosidase and the TUNEL+ cells in tumor tissues.
Reviewer 3 Report
Comments and Suggestions for Authors
This study primarily reports the senescence-inducing effects of the vascular endothelial growth factor receptor inhibitor axitinib on human lung cancer cells and explores the therapeutic effects of the senolytic agent ABT-263 on axitinib-induced senescent lung cancer cells. The senolytic effect of ABT-263 on senescent A549 cells is attributed to caspase-dependent apoptosis and inhibition of Bcl-xL. Reactive oxygen species (ROS) are involved in axitinib-induced senescence in A549 cells but are not involved in senescence clearance. The concept is relatively clear, but the following issues remain:
1. The authors did not knockout the vascular endothelial growth factor receptor (VEGFR) to verify whether axitinib and other drugs exert their senescence-promoting effects independently of VEGFR. Instead, they relied on changes in downstream signaling pathways, which may not provide sufficient evidence.
2. The study mentions that the cells are not very sensitive to lenvatinib itself. Is the concentration used for pre-treating the cells sufficient to induce senescence?
3. A549 and PC9 cells have different molecular characteristics, such as P53. The A549 and PC9 cells were highly and moderately sensitive to axitinib. Therefore, whether axitinib induces senescence through the P53 pathway needs further verification.
4. The axitinib / ABT-263 treatment in mouse experiments caused weight loss. Are there systemic toxic effects? It is recommended to assess whether organs such as the heart, liver, spleen, lungs, and kidneys exhibit senescence or other damage to determine the safety of the drugs.
Comments on the Quality of English Language
The quality of English is good.
Author Response
Reviewer 3
Thank you very much for taking the time to review our manuscript. Please find the detailed responses below and the corresponding revisions highlighted (in red) in the re-submitted files.
1) The authors did not knockout the vascular endothelial growth factor receptor (VEGFR) to verify whether axitinib and other drugs exert their senescence-promoting effects independently of VEGFR. Instead, they relied on changes in downstream signaling pathways, which may not provide sufficient evidence.
Response 1: Thank you for important comments. We at first examined the expression of VEGFRs on a panel of human cancer cell lines (Fig. 1D and Fig. 2B). Although the expression levels of VEGFRs were varied and some cancer cells did not express these receptors on their surface, axitinib induced senescence in all cancer cell lines. In addition, we performed immunoblot to examine effects of axitinib or lenvatinib on phosphorylation of Akt and Erk1/2, as downstream molecules of VEGFR signaling, but no clear results were obtained. Based on these results, we suppose that axitinib induced senescence in cancer cells via unknown mechanism other than its TKI effect on VEGFR signaling, as described on page 15, line 462-468. We are sorry that we will perform knockout experiments because the allowed period of revision is only 7 days. On the other hand, we recently reported that pemetrexed, a folate inhibitor, can induce senescence in A549 and PC9 cells (ref. 26).
2) The study mentions that the cells are not very sensitive to lenvatinib itself. Is the concentration used for pre-treating the cells sufficient to induce senescence?
Response 2: Thank you for comments. When cancer cells were culture at higher doses, i.e., 40 uM, of lenvatinib, cancer cells were dead. The dose of 2.5 uM of lenvatinib showed no effect on cancer cells, whereas this dose of axitinib induced senescence clearly. This is the novelty of this study.
3) A549 and PC9 cells have different molecular characteristics, such as P53. The A549 and PC9 cells were highly and moderately sensitive to axitinib. Therefore, whether axitinib induces senescence through the P53 pathway needs further verification.
Response 3: Thank you for insightful comments. That is an important point. A549 and MCF-7 cell lines have wild type p53, but PC9 and MDA-MB-231 cell lines carry mutant p53. Nevertheless, axitinib induced senescence in all cell lines. This means that p53 was unrelated to axitinib-induced senescence. Interestingly, it has been reported that SASP, a feature of senescence, is limited by wild type p53 [35]. Alternatively, ABT-263-induced senolysis was higher in p53 wild type A549 cells than in p53 mutated PC9 cells. At present, we suppose that p53 plays an important role in senolysis but not in the induction of senescence. These are described on page 16, line 493-500.
4) The axitinib / ABT-263 treatment in mouse experiments caused weight loss. Are there systemic toxic effects? It is recommended to assess whether organs such as the heart, liver, spleen, lungs, and kidneys exhibit senescence or other damage to determine the safety of the drugs.
Response 4: Thank you for important comments. To evaluate adverse events, we measure the body weight of cancer-bearing and treated mice. In vivo administration of higher doses of either or both of axitinib and ABT-263 resulted in toxicity. The reviewer suggests to assess organs on treated mice to evaluate toxicity. However, it is impossible for us to do such experiments because the allowed period of revision is only 7 days. We would like to do such experiments in the next study.
Round 2
Reviewer 1 Report
Comments and Suggestions for Authors
The authors performed corrections accurately and recommend publishing.
Reviewer 3 Report
Comments and Suggestions for Authors
The authors have solved all my concerns
Comments on the Quality of English LanguageNo comments